# Instructive Dialogue Summarization with Query Aggregations

**Bin Wang,  Zhengyuan Liu,  Nancy F. Chen**
Institute for Infocomm Research (I²R), A*STAR, Singapore
wang_bin@i2r.a-star.edu.sg

## Abstract

Conventional dialogue summarization methods directly generate summaries and do not consider user's specific interests. This poses challenges in cases where the users are more focused on particular topics or aspects. With the advancement of instruction-finetuned language models, we introduce instruction-tuning to dialogues to expand the capability set of dialogue summarization models. To overcome the scarcity of instructive dialogue summarization data, we propose a three-step approach to synthesize high-quality query-based summarization triples. This process involves summary-anchored query generation, query filtering and query-based summary generation. By training a unified model called ***InstructDS*** (**Instruct**ive **D**ialogue **S**ummarization) on three summarization datasets with multi-purpose instructive triples, we expand the capability of dialogue summarization models. We evaluate our method on four datasets, including dialogue summarization and dialogue reading comprehension. Experimental results show that our approach outperforms the state-of-the-art models and even models with larger sizes. Additionally, our model exhibits higher generalizability and faithfulness, as confirmed by human subjective evaluations.

## 1 Introduction

Both Verbal and non-verbal conversations play a crucial role in the realm of communication. They serve as channels for humans to exchange information, ideas and emotions (Kester, 2004). In an era of information explosion overload, dialogue summarization has become increasingly essential. The process involves extracting the key dialogue information, enabling people to grasp the essence of their daily interactions.

Conventional dialogue summarization models typically approach the problem as an unconstrained sequence-to-sequence task, treating

> **Benjamin:** Hey guys, what are we doing with the keys today?
> **Hilary:** I've got them. Whoever wants them can meet me at lunchtime or after
> **Elliot:** I'm ok. We're meeting for the drinks in the evening anyway and
> ...
> **Benjamin:** Interesting 😃 😱 To be honest, Hilary, I almost feel like changing my mind. Wanting to take this nap might end up costing me to dear
> ...
> **Hilary:** Do join us, we're going to have fun. And then you'll take the keys and take this most deserved of naps
> **Elliot:** Sounds like a plan
> **Hilary:** 😃 😎
> **Elliot:** See you at 2 then xx

> **Instruction:** Where is Hilary having lunch?
> **Output:** La Cantina

> **Instruction:** Highlight the key takeaways from the dialogue.
> **Output:** Hilary has the keys to the apartment. Benjamin wants to get them and go take a nap. Hilary is having lunch with some French people at La Cantina. Hilary ...

> **Instruction:** Summarize the dialogue with about 15 words.
> **Output:** Benjamin, Hilary and Elliot are discussing about their plans for the day after getting the apartment keys.

Figure 1: Instructive dialogue summarization models, such as InstructDS, demonstrate multiple capabilities.

dialogue-summary pairs as straightforward input-output pairs (Shang et al., 2018; Goo and Chen, 2018; Chen and Yang, 2020). Although fine-tuning pre-trained language models such as BART (Lewis et al., 2020) has shown promising results, these models fail to consider the specific preferences of users who have distinct backgrounds, objectives, intents, and applications for the summaries they require.

In order to address this challenge, several methods have been proposed to integrate queries when generating summaries (Dang, 2006; Nema et al., 2017; Su et al., 2021; Zhong et al., 2021; Zhu et al., 2022; He et al., 2022). However, these models primarily concentrate on domains such as news (Dang, 2006; He et al., 2022), Wikipedia (Zhu et al., 2022), and meetings (Zhong et al., 2021). The exploration of query-based summarization for

dialogues remains limited. Furthermore, Liu and Chen (2021) propose controllable generation using personal named entity planning, and Wang et al. (2022a) suggest controlling summary conciseness. However, both methods focus on specific aspects of controllability and still lack the flexibility to incorporate user requirements as shown in Figure 1.

The primary obstacle in instruction-based dialogue summarization is the scarcity of training data. While existing datasets contain dialogue-summary pairs, creating query-based dialogue summarization datasets with limited human involvement is challenging due to high costs, limited diversity, and potential quality issues. In this work, shed light by Self-Instruct (Wang et al., 2022c), we propose to synthesize query-dialogue-summary (QDS) triples by leveraging the conditional question generation and answering ability of general large language models (LLMs) (Wei et al., 2023). The process involves requesting LLMs to generate multiple candidate queries based on the reference summary. A filtering mechanism, employing text-based and semantic-based methods, is then applied to ensure the quality of collected queries. Finally, the query-based summarization is generated by triggering the question-answering ability of LLMs. This approach demonstrates a promising solution to generate query-based dialogue summarization triples while reducing human involvement and enhancing data diversity.

The InstructDS framework is shown in Figure 2. Through joint training with QDS triples, InstructDS can cater to user preferences by producing query-based summaries. This mixed training paradigm enhances the model's understanding of dialogue, which improves the factual quality of generated summaries. Our model exhibits superior domain generalizability by incorporating multiple datasets into a unified framework and the user's conciseness requirement can be fulfilled by our length-aware augmentations.

Our main contributions are summarized as follows:

- We introduce **InstructDS**, the pioneering instruction-following dialogue summarization model. It is a text generation model designed to summarize dialogues while explicitly considering user instructions.

- We present a straightforward yet effective approach to synthesize query-dialogue-summary

triples from dialogue-summary pairs, facilitating query-based dialogue summarization. This method leverages the question generation and answering capabilities of large language models (LLMs). We validate its effectiveness through evaluations conducted by human annotators.

- We conducted an extensive evaluation on 3 dialogue summarization datasets and 1 dialogue comprehension dataset. The results demonstrate a substantial improvement over previous models. Additionally, according to human subjective test, our generated summaries exhibit comparable levels of factuality, fluency, informativeness, and conciseness to human written ones.

## 2 Related Work

### 2.1 Dialogue Summarization

Dialogue summarization is the task of generating a concise and fluent summary of a conversation involving two or more participants. It has gained significant attention due to its broad applications and availability of relevant datasets (Gliwa et al., 2019; Chen et al., 2021; Zhao et al., 2021). Solutions on dialogue summarization are mainly based on sequence-to-sequence models including the pointer-generation network (See et al., 2017), T5 (Raffel et al., 2020) and BART (Lewis et al., 2020). However, it remains a challenging task due to the lengthy and unstructured nature of dialogues. Chen and Yang (2020) proposes extracting dialogue structures from various perspectives before summarization. Other approaches attempt to incorporate co-reference information (Liu et al., 2021b) and leverage dialogue understanding objectives (Liu et al., 2021a) to enhance the factuality and informativeness (Tang et al., 2022; Wang et al., 2022b).

Similar to text summarization, the process of generating dialogue summarization is uncontrollable and poses challenges in incorporating user preferences (Zhong et al., 2021; He et al., 2022). Efforts have been made to enhance the controllability of dialogue summarization. However, these approaches often have limited focus on personal named entities (Liu and Chen, 2021, 2022; Wang et al., 2022a) and conciseness (Wang et al., 2022a). The primary challenge in instructive dialogue summarization lies in the availability of suitable su-

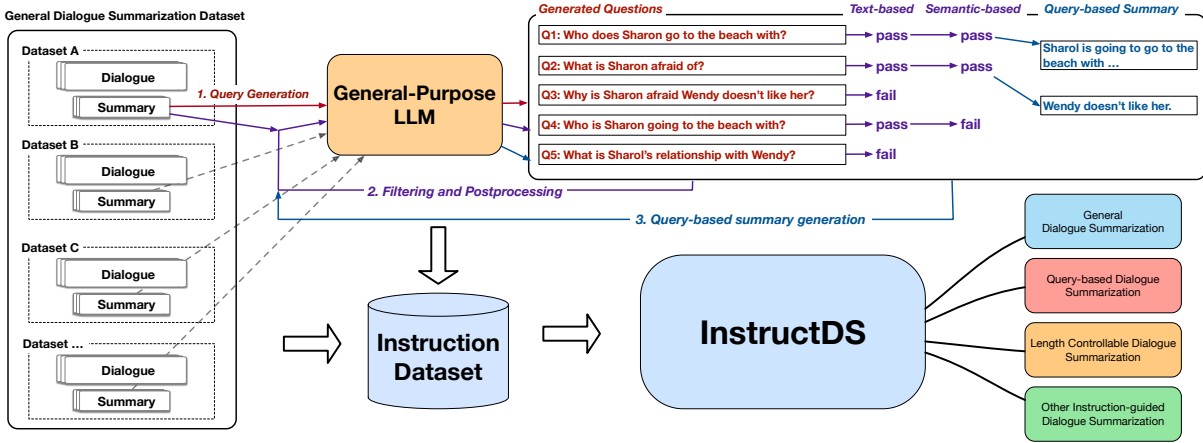

Figure 2: Overall framework of our Instructive Dialogue Summarization (InstructDS) model.

pervision. While QMSum (Zhong et al., 2021) introduces the first query-based meeting summarization, it focuses on lengthy meetings and consists of only 232 meeting samples. To address this limitation, we propose a methodology for synthesizing query-dialogue-summary triples leveraging summary-anchored techniques, to facilitate instructive dialogue summarization.

## 2.2 Instruction Tuning

Recently, instruction-finetuning on large language models has demonstrated remarkable generalizability towards unseen tasks by leveraging task descriptions (Brown et al., 2020; Wang et al., 2022d; Chung et al., 2022). The availability of high-quality and diverse instructions unlocks the emerging capabilities of LLMs. For instance, Flan-series models are tuned with over 1800 tasks with diverse instructions (Chung et al., 2022). However, dialogue tasks, being a sub-domain, have limited supervised data. This limitation leads to sub-optimal performance on query-based dialogue summarization using existing instruction-finetuned models. To mitigate the reliance on human-annotated instruction data, Self-Instruct (Wang et al., 2022c) uses GPT3 for generating diverse instructions and input-output pairs. In a similar vein, our study introduces diverse and high-quality augmentations of query-based dialogue summarization data for instructive dialogue summarization.

## 3 Instructive Dialogue Summarization

### 3.1 Problem Definition

Unlike previous dialogue summarization approaches, instructive dialogue summarization involves a controllable generation process where the summary is dependent on both query and dialogue. In non-instructive dialogue summarization, given dialogue-summary pair $\{\boldsymbol{D}_{d,i}, \boldsymbol{S}_{d,i}\}_{i=1}^{p_d}$ from dataset $d$ with $p_d \geq 1$ pairs of instances, a model $\mathbf{M}$ is expected to generate a summary given the corresponding dialogue: $\mathbf{M}(\boldsymbol{D}_{d,i}) = \boldsymbol{S}_{d,i}$. In instructive dialogue summarization, the focus shifts to structured triples. Given a query-dialogue-summary (QDS) triple $\{\boldsymbol{Q}_d, \boldsymbol{D}_d, \boldsymbol{S}_d\}_{i=1}^{t_d}$ from dataset $d$ with $t_d$ triples, an instructive model $\boldsymbol{M}$ should generate summary conditioned on both query and dialogue: $\boldsymbol{M}(\boldsymbol{Q}_{d,i}, \boldsymbol{D}_{d,i}) = \boldsymbol{S}_{d,i}$. Note that the same dialogue can be shared in several QDS triples with different queries.

### 3.2 Synthesize QDS Triples

The process of generating query-dialogue-summary (QDS) triples from dialogue-summary pairs in our pipeline involves three steps: 1) query generation from complete summary; 2) query filtering to ensure the validity and diversity; 3) query-guided summary generation.

**Query Generation.** In order to capture a diverse range of potential queries, we deploy the question-generation ability of LLMs to generate multiple candidate queries. Specifically, we use *Flan-T5-XL* (Chung et al., 2022) (refer as model $\boldsymbol{X}$) which has been trained on question generation datasets such as Quoref (Dasigi et al., 2019), MC-TACO (Zhou et al., 2019) and CosmosQA (Huang et al., 2019). We expect that its question-generation ability can be generalized to other narrative text. For each instance, we generate five candidate queries using the template shown in Table 7.

**Filtering and Postprocessing.** To ensure validity and diversity, we present two methods for

| Dataset | # Train | # Validation | # Test | # QDS Triples | Direct Exposure | | |
|---------|---------|--------------|--------|---------------|------|-----|-----|
| | | | | | *Alpaca* | *Flan-Series* | *InstructDS* |
| SAMSum (Gliwa et al., 2019) | 14,732 | 818 | 819 | 18,245 | ✗ | ✓ | ✓ |
| DialogSum (Chen et al., 2021) | 12,460 | 500 | 1,500 | 18,600 | ✗ | ✗ | ✓ |
| TODSum (Zhao et al., 2021) | 7,892 | 999 | 999 | 8,705 | ✗ | ✗ | ✓ |
| DREAM (Sun et al., 2019) | 6,116 | 2,040 | 2,041 | - | ✗ | ✓ | ✗ |

Table 1: The dataset statistics include several dialogue summarization datasets such as SAMSum, DialogSum, and TODSum, as well as the DREAM dataset, which focuses on dialogue reading comprehension and contains natural query-dialogue-summary triples. The right part indicates the direct supervision exposures for Alpaca, Flan-Series, and InstructDS models.

| Quality Review Question | Yes% (w/o filtering) |
|-------------------------|----------------------|
| Does the query question answerable? | 94% (76%) |
| Is the query differs from previous ones for the same dialogue? | 90% (63%) |
| Is the generated summary correct and acceptable for query and dialogue? | 83% (71%) |
| Both unique and correct. | 75% (45%) |

Table 2: Quality review for the generated queries and summaries from synthesized DQS triples. After (before) filtering results are shown. Examples of both kept and filtered QDS triples can be found in Table 10, 11. 12.

query filtering. 1) **Text-based filtering.** Through an analysis of candidate queries, we observe that some queries are not answerable conclusively without hallucinations. Examples of such queries include *'What will'* and *'How would'* queries. Therefore, we utilize model $X$ as a text-based binary classifier to determine the answerability of queries using the template in Table 7. This filtering process eliminates around 45% of generated queries that are likely to be unanswerable. 2) **Semantic-based filtering.** To avoid redundancy and ensure diversity, we remove similar queries for the same dialogue-summary pair through semantic similarity measurement. For instance, queries such as *'What does Edward think about Bella?'* and *'What does Edward think of Bella?'* are almost identical in meaning. We keep only the first query if the semantic similarity score, computed using normalized BERTScore (Zhang et al., 2020), is above 0.65. The semantic-based filtering process eliminates an additional 50% of the queries.

**Query-based Summary Generation.** Using the query and the complete summary as input, we generate the query-based summary with model $X$. It is worth noting that generating query-based summaries from dialogues is challenging for model $X$. In contrast, generating query-based summaries from condensed summaries is comparatively easier as it allows the model to extract information from a more concise and structured source, which further guarantees the quality.

**Quality Check.** Finally, we collect QDS triples for three dialogue summarization datasets and present statistics in Table 1. On average, 1.3 QDS triples are generated for each dialogue-summary pair. To access quality and diversity, we enlist help from an expert to annotate 100 triples. Evaluation results in Table 2 demonstrate a significant improvement in the quality of synthesized triples after applying our filtering technique, with the quality score increasing from 45% to 75%. In the process, we incorporate the summary information as it represents a condense version of the information contained in the corresponding dialogues. The triples gathered are tend to have higher quality with fewer errors and cover more utterances as the comparison shown in Table 10.

### 3.3 Model Training

We perform instruction tuning with *Flan-T5-XL* model as the initial checkpoint. The instructions are three-fold: 1) general dialogue summarization, 2) query-based dialogue summarization and 3) their length-aware augmentations. For query-based dialogue summarization, the query and dialogue are concatenated as the input with the template: "*###Instruction: {instruction}. ### Input: {dialogue}.*", where output is the summary. To account for length-aware generations, we append "*The generated summary should be around {summary length} words long.*" to the original instruction.

To enhance the generalizability across different dialogue types, we combine three dialogue summarization datasets to train a unified dialogue summarization model. From the synthesized QDS triples, we random sample 5k triples from each dataset. For length awareness, each sample is augmented once. This results in a total of $(14.7k + 12.5k + $

| Models | Params | ROUGE-1 | | | ROUGE-2 | | | ROUGE-L | | | BS |
|---|---|---|---|---|---|---|---|---|---|---|---|
| | | $F_1$ | *Pre* | *Rec* | $F_1$ | *Pre* | *Rec* | $F_1$ | *Pre* | *Rec* | |
| *Pointer-Generator* | - | 40.1 | - | - | 15.3 | - | - | 36.6 | - | - | - |
| *BART* | 400M | 53.0 | 59.0 | 52.8 | 28.4 | 32.1 | 28.2 | 44.2 | 49.3 | 44.0 | 53.3 |
| *MV-BART* | 400M | 53.9 | 55.7 | 57.4 | 28.4 | 29.3 | 30.6 | 44.4 | 45.7 | 47.5 | 53.6 |
| *Coref-BART* | 400M | 53.7 | 56.9 | 56.4 | 28.5 | 30.5 | 29.7 | 44.3 | 46.9 | 46.5 | 53.5 |
| *ConDigSum* | 400M | 54.3 | 56.0 | 57.6 | 29.3 | 30.4 | 31.2 | 45.2 | 46.6 | 48.0 | 54.0 |
| *GPT-3-finetune* | 175B* | 53.4 | - | - | 29.8 | - | - | 45.9 | - | - | - |
| *Alpaca* | 7B | 28.2 | 26.0 | 39.8 | 5.7 | 5.1 | 8.3 | 20.5 | 19.2 | 29.0 | 19.4 |
| *Flan-T5-XXL* | 11B | 52.6 | 62.6 | 50.0 | 28.5 | 34.1 | 27.1 | 44.1 | 52.5 | 41.9 | 53.2 |
| *Flan-UL2* | 20B | 53.3 | 60.3 | 52.5 | 28.0 | 32.0 | 27.7 | 44.1 | 50.0 | 43.3 | 53.5 |
| *ChatGPT* | 175B | 32.7 | 22.4 | 70.2 | 12.3 | 8.4 | 27.1 | 24.7 | 16.9 | 53.6 | 32.5 |
| *InstructDS* | 3B* | **55.3** | 58.8 | 57.5 | **31.3** | 33.5 | 32.6 | **46.7** | 49.7 | 48.6 | **55.5** |
| *w/ reference summary length* | | | | | | | | | | | |
| *ChatGPT* | 175B | 40.8 | 39.3 | 43.4 | 13.7 | 13.2 | 14.6 | 31.5 | 30.5 | 33.4 | 40.0 |
| *InstructDS* | 3B* | **58.4** | 58.5 | 58.8 | **32.8** | 32.9 | 33.0 | **48.9** | 49.0 | 49.2 | **58.5** |

Table 3: ROUGE scores on SAMSum test set. The results are divided into two blocks: dedicated dialogue summarization models and general-purpose LLMs. "*w/ reference summary length*" indicates reference summary lengths are provided in instructions. * indicates 37.7M trainable parameters.

$7.9k + 5k \times 3) \times 2 = 100k$ samples for training. It is important to note that all experimental results are obtained using the unified model without any specific tuning for individual datasets, which can potentially yield better results but at the cost of reduced generalizability. We employ LORA for parameter efficient training with a total of 37.7 million trainable parameters (Hu et al., 2022).

## 4 Experiments

### 4.1 Datasets, Metrics, and Baselines

We evaluate and benchmark our method on three dialogue summarization datasets including SAM-Sum (Gliwa et al., 2019), DialogSum (Chen et al., 2021) and TODSum (Zhao et al., 2021). These datasets are equipped with dialogues and human-written or verified summaries. Additionally, we explore dialogue reading comprehension with DREAM dataset (Sun et al., 2019) and we evaluate model accuracy without candidate choice exposures. In other words, we reframe DREAM as an open question-answering generation problem. The generated output is subsequently combined with BERTScore to determine the most possible choice. More details and examples are explained in Section A.3. For evaluation metrics, we include ROUGE[1] (Lin, 2004), BERTScore (Zhang

---

[1]We utilize the ROUGE-score package from Google and compared with other implementations in Section A.1.

et al., 2020), human evaluation and ChatGPT (GPT-3.5-Turbo-0301) for comprehensive quality assessment.

Dialogue summarization can be approached using dedicated dialogue summarization models, which are specifically finetuned with in-domain data, as well as general-purpose large language models (LLMs) that have larger model sizes and can perform various tasks by following instructions. The selected benchmarking models are **Dialogue Summarization Model.** 1) Pointer-Generator (See et al., 2017), 2) BART (Lewis et al., 2020), 3) MV-BART (Chen and Yang, 2020), 4) Coref-BART (Liu et al., 2021b), 5) ConDig-Sum (Liu et al., 2021a), 6) GPT-3-finetuned (Hu et al., 2022). **General LLMs.** 7) FLAN-T5 (Chung et al., 2022), 8) FLAN-UL2 (Tay et al., 2022), 9) ALPACA, and 10) ChatGPT.

### 4.2 Main Results

The performance of different models on the SAM-Sum dataset is presented in Table 3, providing insights into their capabilities for general dialogue summarization. Notably, InstructDS outperforms others and establish the new SOTA for both ROUGE and BERTScore metrics. In general, dedicated summarization models show better performance because of their optimization specifically for the single task and dataset. In the case of general LLMs, Alpaca demonstrates less promising

| Models | DialogSum | | | | TODSum | | | |
|---|---|---|---|---|---|---|---|---|
| | R-1 | R-2 | R-L | BS | R-1 | R-2 | R-L | BS |
| *Alpaca* | 25.5 | 4.9 | 18.8 | 18.0 | 33.6 | 6.9 | 21.8 | 14.6 |
| *Flan-T5-Large* | 38.8 | 14.4 | 30.9 | 38.7 | 37.3 | 13.4 | 25.3 | 23.6 |
| *Flan-T5-XXL* | 39.3 | 15.8 | 32.4 | 39.5 | 39.3 | 14.2 | 27.2 | 23.5 |
| *Flan-UL2* | 40.8 | 16.5 | 33.3 | 40.9 | 41.6 | 14.6 | 27.9 | 24.3 |
| *ChatGPT* | 38.4 | 12.9 | 29.8 | 38.8 | 39.8 | 11.8 | 24.5 | 24.9 |
| *BART* | 47.3 | 21.3 | 38.6 | 45.8 | 73.1 | 56.8 | 64.0 | 64.3 |
| *InstructDS* | **47.8** | **22.2** | **39.4** | **47.0** | **89.3** | **78.9** | **85.4** | **85.5** |

Table 4: Results on DialogSum and TODSum dataset. BART results are computed from the outputs released by Chen et al. (2021) and Zhao et al. (2021).

| Models | Multi-Choice Acc. |
|---|---|
| *Random* | 33.3% |
| *Alpaca* | 51.3% |
| *Flan-T5-Large* | 53.1% |
| *Flan-T5-XXL* | 58.5% |
| *Flan-UL2* | 56.8% |
| *ChatGPT* | **60.8%** |
| *InstructDS* | 57.8% |
| + In-domain | **65.9%** |

Table 5: Results on DREAM dataset. InstructDS is not trained with DREAM data. "+In-domain" indicates the inclusion of DREAM training data.

performance due to its optimization using synthesized instruction data, with limited involvement of dialogue summarization tasks. In contrast, as depicted in Table 1, FLAN-based models include the SAMSum dataset in their instruction tuning process, resulting in competitive performance. While ChatGPT is renowned for its versatility across various tasks, it is prone to generate lengthy summaries when not constrained by prompts (Qin et al., 2023). Therefore, we further experiment with adding the reference summary length to instructions during summary generation. This approach significantly improves the performance of ChatGPT, achieving a balance between precision and recall. Meantime, InstructDS exhibits further performance boost, demonstrating its ability to follow length instructions.

We conduct experiments on query-based dialogue summarization using the DREAM dataset and the results are presented in Table 5. InstructDS can achieve comparable performance with Flan-T5

and underperform ChatGPT. It is important to note that InstructDS is only directly trained with synthesized QDS triples from other datasets, whereas FLAN-based models are directly trained with the DREAM data. This demonstrates the effectiveness and generalizability of our synthesized triples. Further, we explored the impact of incorporating in-domain DREAM training data into InstructDS. This resulted in a significant performance boost, surpassing ChatGPT by a considerable margin.

To access the generalizability of InstructDS, we present results on DialogSum and TODSum datasets in Table 4. InstructDS maintenances its outstanding performance over other models. A significant gap exists between fine-tuned BART model and general-purpose LLMs. In contrast, InstructDS incorporates multiple data sources and augmented QDS triples. This comprehensive dialogue understanding framework contributes to its superior reasoning abilities across diverse dialogue domains and tasks.

### 4.3 Ablation Study

To provide insights into the effectiveness of InstructDS, we conduct ablation studies on InstructDS variants to answer two fundamental questions: 1) How do augmented QDS triples contribute to general and query-based dialogue summarization? 2) What is the effect of length awareness augmentation on general and length-controllable dialogue summarization?

**Model variants.** We examine five model variants of InstructDS across four evaluation datasets. Each variant is trained with distinct training data: 1) **MixDS**: Mixing three dialogue summarization datasets, 2) **MixDS+QDS**: MixDS and synthesized QDS triples, 3) **MixDS+Length**: MixDS and

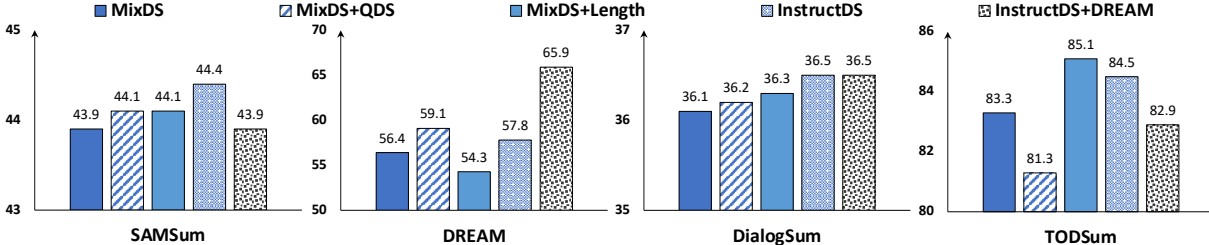

Figure 3: Ablation study on different variants of InstructDS. Averaged ROUGE-1/2/L score is reported for dialogue summarization datasets, including SAMSum, DialogSum and TODSum. Accuracy is computed for DREAM dataset.

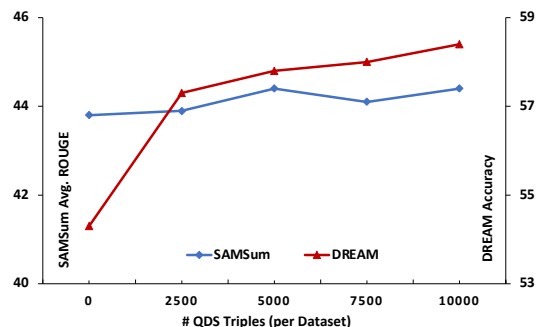

Figure 4: Ablation study on the number of included QDS triples. Performance on SAMSum (left, averaged ROUGE) and DREAM (right, accuracy) datasets are reported.

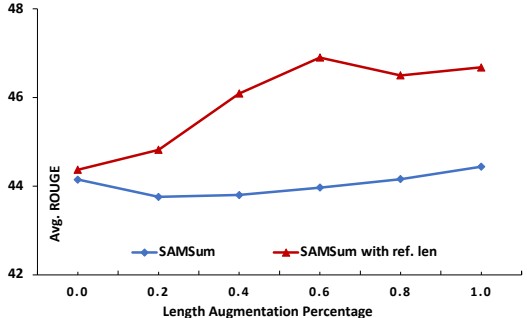

Figure 5: Ablation study on the percentage of length augmented instances. Performance on standard SAMSum and length-revealed SAMSum are reported.

length augmentation, 4) **InstructDS**: MixDS, synthesized QDS triples and length augmentation, 5) **InstructDS+DREAM**: MixDS, synthesized QDS triples, human-written QDS triples from DREAM, and length augmentation. All other hyperparameters and evaluation protocols are kept identical for fair comparisons.

The ablation results are presented in Table 3. First, on SAMSum and DialogSum datasets, the full InstructDS model demonstrates the best performance. Incorporating both synthesized QDS triples and length augmentation techniques contributes to an overall performance improvement. We attribute these performance boosts on the enhanced dialogue understanding and length awareness capabilities. On DREAM dataset, the inclusion of synthesized QDS triples leads to an improvement in query-based dialogue summarization performance, elevating it from 56.4 to 59.1. Notably, the performance is further enhanced when additional in-domain training data from the DREAM dataset. In TODSum dataset, we observe that augmented QDS triples do not yield better summarization results. However, the utilization of length augmentation techniques does improve the performance. It is

because TODSum summaries are more templated, requiring less reasoning. In summary, our findings demonstrate that augmented QDS triples enhance dialogue understanding and reasoning capabilities, enabling query-based dialogue summarization with transferability to other dialogue datasets (such as DREAM). Length augmentation, on the other hand, enables not only length controllability but also improves length awareness in general summarization tasks.

**Number of QDA Triples.** To examine the impact of varying numbers of augmented QDS triples on InstructDS, we conduct experiments on SAMSum and DREAM datasets, as shown in Figure 4. Consistent with previous observations, the inclusion of augmented QDS triples guides the model to reason more effectively, resulting in improved performance on general dialogue summarization. Additionally, the results from the DREAM dataset indicate that increasing the number of synthesized QDS triples leads to better performance in query-based summarization. As the number of QDS triples increases, the accuracy improves from 54 to over 58.

**Length Augmentation Percentage.** The impact of varying the percentage of added length augmen-

| Models | Human Annotator | | | | ChatGPT | | | |
| --- | --- | --- | --- | --- | --- | --- | --- | --- |
| | Faithfulness | Fluency | Informativeness | Conciseness | Faithfulness | Fluency | Informativeness | Conciseness |
| *BART* | $3.85_{(1.3)}$ | $4.36_{(0.8)}$ | $3.22_{(1.0)}$ | $4.30_{(0.9)}$ | $4.22_{(1.1)}$ | $4.80_{(0.5)}$ | $3.37_{(1.0)}$ | $4.93_{(0.3)}$ |
| *Alpaca* | $3.24_{(1.3)}$ | $3.77_{(1.3)}$ | $3.45_{(1.1)}$ | $3.11_{(1.4)}$ | $3.59_{(1.3)}$ | $4.07_{(1.0)}$ | $3.19_{(1.2)}$ | $4.29_{(1.0)}$ |
| *Flan-UL2* | $4.00_{(1.3)}$ | $4.38_{(0.9)}$ | $3.03_{(1.2)}$ | $4.29_{(1.0)}$ | $4.45_{(0.9)}$ | $4.78_{(0.5)}$ | $3.52_{(1.0)}$ | $4.91_{(0.3)}$ |
| *ChatGPT* | $4.52_{(0.9)}$ | $4.38_{(0.9)}$ | $4.62_{(0.6)}$ | $2.77_{(1.4)}$ | $4.94_{(0.3)}$ | $4.94_{(0.2)}$ | $4.78_{(0.4)}$ | $4.89_{(0.3)}$ |
| *Human-written* | $4.34_{(1.0)}$ | $4.54_{(0.7)}$ | $3.58_{(1.1)}$ | $4.36_{(0.9)}$ | $4.49_{(0.8)}$ | $4.81_{(0.4)}$ | $3.74_{(1.0)}$ | $4.95_{(0.3)}$ |
| *InstructDS* | $4.13_{(1.1)}$ | $4.35_{(0.8)}$ | $3.54_{(1.0)}$ | $4.23_{(1.0)}$ | $4.60_{(0.8)}$ | $4.82_{(0.4)}$ | $3.78_{(0.9)}$ | $4.92_{(0.3)}$ |

Table 6: Subjective quality evaluation with instances random sampled from SAMSum dataset (30 samples for Human Annotators, 200 samples for ChatGPT.). The Mean and standard deviation of evaluation scores are reported.

tation samples is shown in Figure 5. It reveals that increasing the number of length augmentations can enhance the model's ability to control the generated summary length while a saturation point exists. Meantime, the effect on general summarization is diverse and relatively less consistent.

## 4.4 Subjective Quality Evaluation

We conduct multidimensional evaluations to access the quality of generated summaries. This involves fine-grained Likert scores (scale 1 to 5, the higher the better) from both human and ChatGPT in four dimensions: Faithfulness, Fluency, Informativeness, and Conciseness (Wang et al., 2022b; Gao et al., 2023a). Evaluations are performed on SAMSum dataset and we randomly sampled 30 instances for human evaluation and 200 instances for ChatGPT evaluation. The user interface and prompt template can be found in Figure 6 and Table 7, respectively. Each dialogue was accompanied by one human-written summary and five machine-generated ones. We engaged 12 volunteers, resulting in 792 labeled samples. On average, each dialogue-summary pair receives assessments from 4.4 annotators. The mean and standard deviation of Likert scores are presented in Table 6.

With human annotations, fluency is the best-performing metric. All models, except for Alpaca, demonstrate the ability to generate fluent summaries. Alpaca's relatively poor performance can be attributed to its unsupervised training and limited exposure to dialogue data. For informativeness and conciseness, ChatGPT and Alpaca produce the most informative summaries but receive the lowest conciseness scores. These models tend to generate longer summaries, including elaborate details, indicating a limited understanding of the desired compressiveness. Faithfulness evaluation emerges as a crucial factor in practical applications, where ChatGPT surpasses human performance. This can be attributed to potential inaccuracies in the annota-

tions of the SAMSum dataset (Wang et al., 2022b; Gao et al., 2023b). ChatGPT's ability to generate detailed content, similar to the concept of Chain-Of-Thought (Wei et al., 2022), also contributes to higher faithfulness. Overall, InstructDS achieves comparable performance to human-written summaries in terms of fluency, informativeness, and conciseness. While InstructDS still falls short on human-level faithfulness, it demonstrates noticeable improvements compared to previous models.

When using ChatGPT as an off-the-shelf evaluator, we observe that InstructDS is achieving on-pair or better performance over human written summaries on four dimensions. Especially for faithfulness, InstructDS is outperforming all other models except for ChatGPT. However, it is worth noting that ChatGPT exhibits biases towards its own outputs, resulting in potentially inflated evaluation scores. Similar patterns have also been found in other studies that involve ChatGPT evaluation. For example, Zhang et al. (2023) shows that ChatGPT always assigns higher scores to its own outputs when solving math problems, leading to significant concerns when using ChatGPT as an evaluator. Furthermore, in this work, we found that ChatGPT is not effective in evaluating the conciseness of summaries, which introduces a noticeable discrepancy compared to human evaluators in this aspect. We think it is because ChatGPT is not aware of the desired conciseness of summaries, which also attributes to the phenomenon that it is generating lengthy summaries. Further explorations are necessary for a robust ChatGPT evaluator in dialogue and other domains (Wang et al., 2023b).

## 5 Discussions

### 5.1 Relationship with Query-focused Summarization

Query-focused summarization is closely related to our instructive dialogue summarization con-

cept (Vig et al., 2022). There are some similarities and differences. An ideal instructive dialogue summarization model should be capable of handling a wide range of instructions when generating summaries. As illustrated in Figure 2, our current model can accommodate general dialogue summarization, query-based dialogue summarization, and dialogue summarization with length control. We anticipate that the range of instructions will be expanded in future research, encompassing diverse sets of instructions and multi-round dialogue summarization scenarios. In the meantime, as a domain-specific model, we anticipate that instructive dialogue summarization could exhibit emergent capabilities as shown in general instruction-tuned LLMs.

### 5.2 Long Dialogue (Meeting) Summarization

Summarizing dialogues in meetings, especially long ones, is a challenging task that requires a model capable of processing extended sequences. One promising avenue for research involves expanding our current method for summarizing lengthy dialogues, which can improve query-based meeting summarization and comprehension. Instead of relying on the entire lengthy dialogue, our approach generates queries from reference summaries. This approach addresses the difficulties of using pre-trained language models for long dialogue inputs in meeting scenarios. One of the obstacles in meeting summarization is the limited data availability, which limits the model's ability to generalize across different domains. Nevertheless, our method has the potential to alleviate data scarcity issues in the context of summarizing lengthy meetings. Another challenge is that meeting summarization is often associated with transcripts with ASR errors (Jiang et al., 2023), which effect is under-explored in existing research.

## 6   Conclusion

In this work, we present InstructDS, which is the first instructive dialogue summarization model that excels in both general dialogue summarization and query-based dialogue summarization. InstructDS can generate high-quality query-based summarization tailored to user requirements. This achievement is made possible through a combination of multi-dataset training, synthesized QDS triples and length-awareness augmentations. Experimental results demonstrate that InstructDS establishes new state-of-the-art performance across all four benchmark datasets.

## Limitations

Dialogue summarization is a label-intensive task that demands substantial supervision and the collection of human-written summaries, which is both challenging and resource-intensive. Moreover, the transferability of annotations across different dialogue domains introduces additional complexity. Therefore, to develop a highly adaptable dialogue summarization model, leveraging unsupervised dialogue data becomes crucial. However, it is worth noting that InstructDS does not incorporate unlabelled dialogue data, leaving room for potential improvement.

Another important aspect to consider in dialogue data is privacy. The sensitive nature of dialogues can hinder the accessibility and public availability of diverse dialogue datasets. Therefore, future enhancements of InstructDS should address privacy concerns and explore the utilization of advanced learning techniques such as federated learning, which can enable collaborative and privacy-preserving training processes.

Automatically evaluating the quality of dialogue summarization poses significant challenges. Acquiring human annotators for model development is expensive and inefficient. Existing evaluation metrics heavily rely on ROUGE, and ChatGPT has emerged as a newly proposed method for evaluation. As discussed in Section 4.4, it still lacks transparency and robustness. Therefore, there is a pressing need for more effective evaluation methods specifically tailored for dialogue summarization. Multilingual and multicultural evaluation is crucial since dialogues are frequently intertwined with local norms, slang, code-switches and cultural nuances (Wang et al., 2023a).

## Acknowledgement

This research was supported by funding from the Institute for Infocomm Research (I$^2$R), A*STAR, Singapore. The computational work for this article was partially performed on resources of the National Supercomputing Centre (NSCC), Singapore.

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

## A  Datasets and Metrics

In this section, we will provide a detailed explanation of the datasets and metrics that have been utilized in our work.

### A.1  ROUGE Implementation

In the context of dialogue summarization, we observed that different studies incorporate different versions of the ROUGE metric, resulting in varied results. Specifically, we identified three widely used implementations of ROUGE:

- ROUGE-1.5.5: It is released by the author of the ROUGE paper. However, this implementation has certain limitations. It requires running the Perl script, which can be inconvenient when integrating it with other projects. Additionally, the last update for this implementation dates back to 2005.

- Py-rouge: A full Python implementation of ROUGE metric. It produces the same result as the Perl implementation. However, the package is not maintained since 2018. Because several dialogue summarization papers use this package, we also report the results using this package in this appendix.

- ROUGE-score: Another Python implementation of ROUGE. It is released and maintained by the Google research team. Recently, HuggingFace incorporate this package into their evaluation metrics package. Because of the influence of Google research and HuggingFace, we expect the package will receive significant attention across various research domains. Consequently, in this paper, we utilize this implementation to calculate the ROUGE score. It is also actively updated, with the latest update in March 2023.

This section presents the main results using the Py-rouge package. The results for SAMSum dataset are presented in Table 8 while the results for DialogSum and TODSum are in Table 9. These results align with the patterns and conclusions discussed in Section 4.2.

### A.2  DialogSum Preprocessing

In the original DialogSum paper (Chen et al., 2021), the authors used #Person1#, #Person2#, and so on to represent speakers because the original dialogue did not contain speaker information. However, this

approach leads to inconsistencies as some names were already present in the original dialogue. To address this issue, we performed additional preprocessing on the data to align its format with SAM-Sum. Specifically, we employed the prompt template shown in Table 7 to prompt the *FLAN-T5-XL* model to predict the name of the person. We then applied rule-based filtering to determine the appropriateness of the predicted names. This filtering process involved considering factors such as forbidden names labeled by humans, the length of the predicted name, the presence of special symbols, and whether the predicted name has appeared in the original dialogue. If the name did not meet the criteria according to our rule-based identification method, we used *FLAN-T5-XL* again with the template from Table 7 to choose from a pool of ten candidate names, which consisted of five randomly sampled male names and five randomly sampled female names. Simultaneously, the name was correspondingly updated in the reference summary.

Examples of preprocessed dialogues can be found in Table 11 and Table 16. To facilitate future research and development, we will make the name-replaced version available for public access.

### A.3 Evaluation on DREAM

DREAM (Sun et al., 2019) dataset is introduced for dialogue reading comprehension and the evaluation is designed as multi-choice questions. However, in real-world applications, where information queries are performed on dialogues, it is unlikely to have several candidate answers included as input. Real-world scenarios in dialogue reading comprehension are better represented as unconstrained text generation problems.

our evaluation of the DREAM dataset is conducted in an unconstrained manner, without providing candidate choices to the model. To assess accuracy, we utilize the BERTScore package to measure the similarity between the generated output and the answer choices, selecting the highest-scoring option as the final answer. The evaluation process is illustrated in Figure 7 and Figure 8.

### B Templates and Examples

We provide more detailed illustrations of the human evaluation interface, instruction templates, synthesized QDS triples and case studies on model outputs.

- Figure 6: the interface used for subjective

evaluation for annotators. The annotator is asked to label the quality of summaries in four dimensions.

- Table 7: the templates used for several tasks including query generation, text-based filtering, ChatGPT evaluation and the preprocessing on DialogSum dataset.

- Table 10, 11 and 12: shows the synthesized QDS triples including both kept and removed ones. The reason for filtering is also demonstrated.

- Table 13, 14, 15, 16 and 17: shows case studies on the generated summaries and query-based summaries on all four datasets including SAMSum, DREAM, DialogSum, and TODSum.

| Task | Prompt Template |
|---|---|
| **Query Generation** (Sec. 3.2) | Generate an answerable and specific question based on the following context. Context: ${Summary} |
| **Text-based Filtering** (Sec. 3.2) | Can we get an answer from the context, yes or no? Question: ${Question} Context: ${Summary} |
| | Is the question fully answerable from the context without any guessing, yes or no? Question: ${Question} Context: ${Summary} |
| **ChatGPT Evaluation** (Sec. 4.4) | Evaluate the quality of the abstractive summary from the dialogue. Please be extremely picky. Rate each summary on four dimensions: Faithfullness: whether the summary is correct according to dialogue, Fluency: Whether summary is grammarly correct, Informativeness: Whether the summary contains all essential information, Conciseness: Whether the summary is very concise (not verbose). Output should follow the template: 'Faithfulness': value, 'Fluency': value 'Informativeness': value, 'Conciseness': value. You should rate on a scale from 1 (worst) to 5 (best). Do not give detailed explanations. Dialogue ${Dialogue}. Summary: ${Summary} |
| **DialogSum Preprocessing** (Sec. A.2) | (1) Who is #Person1# in the following dialogue? ${Dialogue} (2) Select on proper name for #Person1# from ${candidate names} in the following dialogue? ${Dialogue} |

Table 7: Prompting templates for QDS triple generation, text-based query filtering, ChatGPT evaluation, and DialogSum preprocessing.

| Models | ROUGE-1 | | | ROUGE-2 | | | ROUGE-L | | |
|---|---|---|---|---|---|---|---|---|---|
| | $F_1$ | $Pre$ | $Rec$ | $F_1$ | $Pre$ | $Rec$ | $F_1$ | $Pre$ | $Rec$ |
| *BART* | 53.2 | 59.3 | 53.1 | 28.6 | 32.3 | 28.4 | 50.3 | 54.9 | 49.9 |
| *MV-BART* | 54.0 | 55.8 | 57.5 | 28.5 | 29.5 | 30.7 | 50.6 | 51.5 | 53.2 |
| *Coref-BART* | 53.9 | 57.1 | 56.6 | 28.6 | 30.7 | 29.8 | 50.4 | 52.5 | 52.3 |
| *ConDigSum* | 54.4 | 56.2 | 57.8 | 29.4 | 30.5 | 31.4 | 51.3 | 52.3 | 53.7 |
| *Alpaca* | 28.3 | 26.1 | 40.0 | 5.7 | 5.2 | 8.4 | 26.5 | 24.6 | 34.9 |
| *Flan-T5-Large* | 51.3 | 58.7 | 50.1 | 25.9 | 29.7 | 25.4 | 48.7 | 54.2 | 47.3 |
| *Flan-T5-XXL* | 52.8 | 62.8 | 50.2 | 28.6 | 34.2 | 27.3 | 50.2 | 57.8 | 47.7 |
| *Flan-UL2* | 53.4 | 60.4 | 52.6 | 28.1 | 32.2 | 27.8 | 50.2 | 55.5 | 49.1 |
| *ChatGPT* | 32.9 | 22.5 | 70.6 | 12.4 | 8.5 | 27.3 | 31.6 | 22.4 | 59.3 |
| *InstructDS* | 55.6 | 59.1 | 57.8 | 31.4 | 33.6 | 32.8 | 52.7 | 55.1 | 54.2 |
| *w/ reference summary length* | | | | | | | | | |
| *ChatGPT* | 40.9 | 39.5 | 43.5 | 13.8 | 13.2 | 14.7 | 37.7 | 36.6 | 39.5 |
| *InstructDS* | 58.6 | 58.7 | 59.0 | 33.0 | 33.1 | 33.2 | 54.5 | 54.6 | 54.8 |

Table 8: SAMSum results using Py-rouge package.

| Models | DialogSum | | | TODSum | | |
|---|---|---|---|---|---|---|
| | R-1 | R-2 | R-L | R-1 | R-2 | R-L |
| *Alpaca* | 25.6 | 4.9 | 24.6 | 33.4 | 6.9 | 28.1 |
| *Flan-T5-Large* | 38.7 | 14.3 | 37.2 | 37.2 | 13.3 | 31.6 |
| *Flan-T5-XXL* | 39.3 | 15.8 | 38.7 | 39.2 | 14.1 | 33.7 |
| *Flan-UL2* | 40.7 | 16.5 | 39.6 | 41.5 | 14.6 | 34.4 |
| *ChatGPT* | 38.4 | 12.8 | 36.3 | 39.6 | 11.7 | 30.8 |
| *BART* | 47.2 | 21.1 | 44.8 | 73.1 | 56.8 | 68.7 |
| *InstructDS* | 47.7 | 21.8 | 45.3 | 89.2 | 79.8 | 87.5 |

Table 9: DialogSum and TODSum results using Py-rouge package.

Figure 6: An illustration of the user interface for human evaluation of summarization qualities.

**Dialogue:**

W: Tom, look at your shoes. How dirty they are! You must clean them.

M: Oh, mum, but I cleaned them only yesterday.

W: They are dirty now. You must clean them again.

M: I do not want to clean them today. Even if I clean them today, they will get dirty again tomorrow.

W: All right, then.

M: Mum, give me something to eat, please.

W: You had your breakfast in the morning, Tom, and you had lunch at school.

M: I am hungry again.

W: Oh, hungry? But if I give you something to eat today, you will be hungry again tomorrow.

**Query:**

Why did the woman say that she wouldn't give him anything to eat?

**Candidate Choices:**

A: Because his mother wants to correct his bad habit.

B: Because he had lunch at school.

C: Because his mother wants to leave him hungry.

Seq2Seq Model

**Final Answer:**
Because his mother wants to correct his bad habit.

Figure 7: Constrained DREAM evaluation as multi-choices question answering.

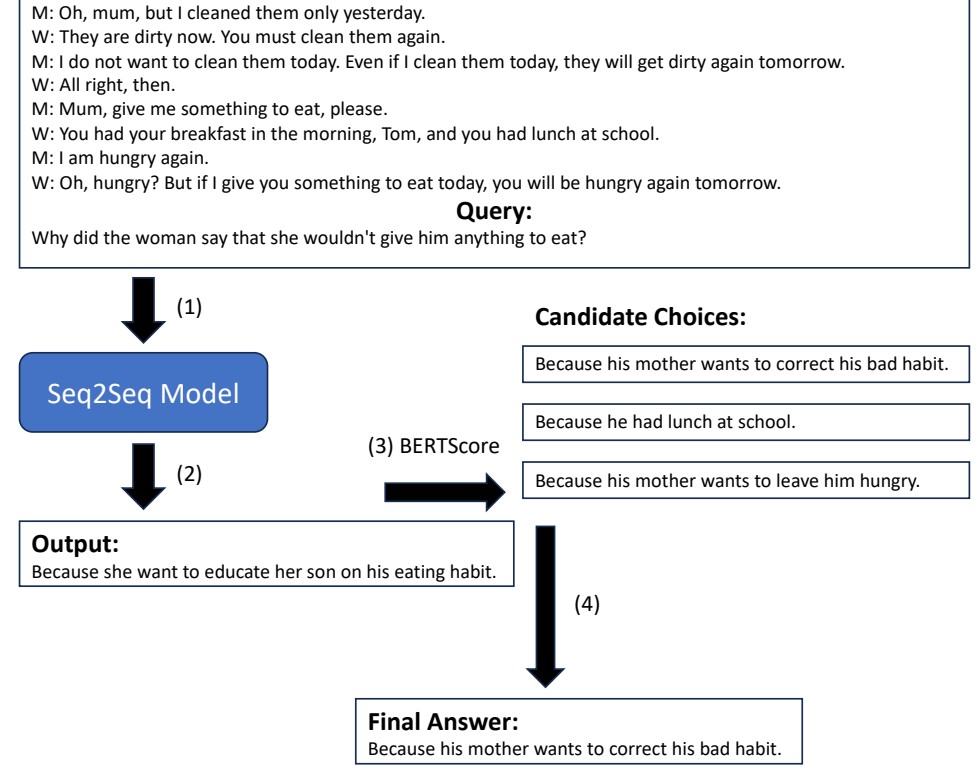

Figure 8: Unconstrained DREAM evaluation as open-ended question answering.

| Dialogue (SAMSum) |
|---|
| Emma: We are going beach would you like to join in? |
| Sharol: sure who else is coming? |
| Emma: its me anna, emily, wendy, kate and you |
| Sharol: would wendy be ok with me joining.. |
| Emma: i think she would be .. is there any problem between you guys? |
| Sharol: i think she doesnt like me ... she always try to avoid me... |
| Emma: really? then i think you should definetly join and sort things with her |
| Sharol: hmm.. i dont want to be her friend forcefully... |
| Emma: i know she doesnt dislike you there must b some misunderstanding... |
|     lets meet up and sort out.. be at my place at 11am |
| Sharol: sure will be there |
| General Summary: Sharol is going to go to the beach with Emma, anna, emily, |
|       wendy and kate. Sharol is afraid that wendy doesn't like her. |

**Dialogue-Guided Question Generation**

| |
|---|
| Query: Sharol will be at Emma's place at 11am. |
| Query: What is Sharol going to do? |
| Query: What is the problem between Sharol and Kate? |
| Query: What is Emma going to do? |
| Query: What will they do next? |

**Kept QDS Triples**

| |
|---|
| Query: Who does Sharol go to the beach with? |
| Summary: Sharol is going to go to the beach with Emma, Anna, Emily, Wendy and Kate. |
| Query: What is Sharol afraid of? |
| Summary: wendy doesn't like her |

**Filtered QDS Triples**

| |
|---|
| Query: Why is Sharol afraid Wendy doesn't like her? |
| Reason for rejection: **text-based filtering**, the answer may not be answerable. |
| Query: Who is Sharol going to the beach with? |
| Reason for rejection: **semantic-based filtering**, duplicate question. |
| Query: What is Sharol's relationship to Wendy? |
| Reason for rejection: **text-based filtering**, the answer may not be answerable. |

Table 10: One example from SAMSum dataset on the synthesized Query-Dialogue-Summary triples.

| Dialogue (DialogSum) |
|---|
| Byron: Are you busy tomorrow night? I'm going over to the Workers Club for volleyball if you'd like to come along. |
| Jacquenette: Isn't that far away? |
| Byron: Not really. If you take the No. 3 bus, you can get there in ten minutes. And if volleyball doesn't interest you, they've got a huge indoor swimming pool, a weight room, and indoor tracks. It's a great place to meet people. Would you like to go with me? |
| Jacquenette: Now that you mentioned it, it would be nice to get away from the computer center for a change, and I really should get some more exercise. Working up a sweat in a ball game or the weight room would be nice. I've got so fat. |
| Byron: Just look at me! You're not the only one. In high school I had a lot more time to do sports activities. Now what little spare time you have must be used in doing exercise. At least the club is open until 12 pm. |
| Jacquenette: I guess it's worth a visit. Did you say you're going tomorrow night? |
| Byron: Yeah. |
| Jacquenette: OK, I'll come, too. How about meeting me in front of the cinema at eight, and we'll catch the bus there. |
| Byron: Sure, see you then. |
| General Summary: Lance invites Daffy to play volleyball together tomorrow night. Daffy realizes it's time for more exercise so Daffy agrees. Lance begins to talk about how much exercise Lance did in high school and how little time Lance can put into exercise now. They decide to meet in front of the cinema. |

**Kept QDS Triples**

| |
|---|
| Query: What do Byron and Jacquenette decide to do? |
| Summary: They think they have been getting too fat and decide to do more sports. |
| Query: What does Byron invite Jacquenette to do tomorrow night? |
| Summary: go over to the Workers Club for volleyball |

**Filtered QDS Triples**

| |
|---|
| Query: What sport do Byron and Jacquenette decide to do? |
| Reason for rejection: **semantic-based filtering**, duplicate question. |
| Query: What sport do Byron and Jacquenette decide to do more of? |
| Reason for rejection: **text-based filtering**, the answer may not be answerable. |
| Query: What does Byron invite Jacquenette to do tomorrow night? |
| Reason for rejection: **semantic-based filtering**, duplicate question. |

Table 11: One example from DialogSum dataset on the synthesized Query-Dialogue-Summary triples.

| Dialogue (TODSum) |
|---|
| User: Can you find me a Chinese restaurant please? |
| Agent: Sure. There are 17 chinese restaurants in town. Are you looking for a certain price range or location? |
| User: Yes. I would like it to be in the expensive price range. |
| Agent: I have found 5 that fit that description. Is there a particular area you would like? |
| User: I would like to book a table for 6 at 15:15 on Tuesday. |
| Agent: I was able to get a table at Hakka in the north. The reference number is 7KW8MVL3 and they'll keep the table for 15 minutes. Does that suit you? |
| User: Yes, thank you. That's all for now. Bye! |
| Agent: Thank you for contacting us and enjoy your meal at the hakka. Have a great day. |
| General Summary: the user wants to book the restaurant seat for 6 people at 14:15 on tuesday. the specialty of the restaurant is chinese food and it is expensive. |

| **Kept QDS Triples** |
|---|
| Query: What is the price range of the restaurant? |
| Summary: the user wants to book the seat for 6 people at 14:15 on tuesday. the specialty of the restaurant is chinese food and it is expensive. |
| Query: What type of food does the restaurant serve? |
| Summary: chinese |
| Query: What type of restaurant is the user interested in? |
| Summary: chinese |

| **Filtered QDS Triples** |
|---|
| Query: What type of food is the restaurant? |
| Reason for rejection: **semantic-based filtering**, duplicate question. |
| Query: What type restaurant is the user looking for? |
| Reason for rejection: **semantic-based filtering**, duplicate question. |

Table 12: One example from TODSum dataset on the synthesized Query-Dialogue-Summary triples.

| Case #1 from SAMSum |
|---|
| **Dialogue:** |
| Marsha: Guys, we've planned the trip with John last night as we promised
Cynthia: great, thank you for that
Marsha: but of course you have to agree on that
Mohammad: sure, but I really trust you
Gavin: me too
Marsha: so as we decided last time, we will spend a week just on the beach
Marsha: we all wanted some calm, nice place, right?
Gavin: yes!!!
Marsha: John found this little hotel in Nosy Be
Marsha: <file_photo>
Marsha: and their website: <file_other>
Marsha: it's quite basic the website, but it may actually be a good sign
Gavin: I love it!
Cynthia: it seems just perfect
Marsha: I know
Marsha: it's more expensive than hotels on the mainland
Marsha: but it seems to be the most beautiful place in Madagascar
Mohammad: so let's go there, it's still cheaper than Italy for example
Marsha: exactly! |
| **Summary - Human Reference** |
| Marsha and John planned the trip. They will spend a week on the beach with Cynthia, Mohammad and Gavin. They will all stay in a hotel in Nosy Be. |
| **Summary - BART** |
| Marsha, Cynthia, Mohammad and Gavin are going to Madagascar. They will spend a week on the beach. John found a hotel in Nosy Be. |
| **Summary - Alpaca** |
| The group agreed to spend their vacation on a beach in the small hotel John had found inNosiv Be, Madgascar. It was more expensivethan hoteles on main land,but it was the best place according to the group. They all agreed that it would be worth the money. |
| **Summary - Flan-UL2** |
| Marsha, Cynthia, Mohammad and Gavin are going to Madagascar. They will spend a week on the beach in Nosy Be. |
| **Summary - ChatGPT** |
| Marsha informs the group that John had planned the trip, and they all agreed to spend a week on the beach. She shows them a picture of a hotel in Nosy Be and its website, which they all like despite it being more expensive than hotels on the mainland. Mohammad mentions that it is still cheaper than Italy and everyone agrees to go there. |
| **Summary - InstructDS** |
| Marsha, John, Cynthia, Mohammad and Gavin will spend a week on the beach in Nosy Be. It's more expensive than hotels on the mainland, but it's still cheaper than Italy. |

Table 13: The first case study from SAMSum dataset and the generated summaries from different models.

| Case #2 from SAMSum |
|---|
| **Dialogue:** |
| Ken: Hi, how are you?
Ang: Just peachy! You?
Ken: I'm okay...
Ang: Just okay? What's wrong?
Ken: Just stressed; work stuff, fighting with Brad, too much going on at mom's.
Ang: Hang in there, it will get better!
Ken: I know, but it's a lot.
Ang: Can I do anything to help?
Ken: You are! Listening to me vent! LOL!
Ang: Are you at least doing anything fun this weekend?
Ken: Show Saturday night, then seeing the grandkids on Sunday at the zoo.
Ang: Sounds great! That will cheer you up!
Ken: Gotta run, work calls. Love you!
Ang: Love you too! Have a fantastic day!
Ken: You too! |
| **Summary - Human Reference** |
| Ken feels stressed because of work and fighting with Brad. There is also too much going on at mom's. Ken is going to a show on Saturday night. On Sunday Ken is seeing the grandkids at the zoo. |
| **Summary - BART** |
| Ken is stressed, because he has to deal with work issues and family issues. He is going to see a show on Saturday night and see the grandkids on Sunday. |
| **Summary - Alpaca** |
| <noinput> |
| **Summary - Flan-UL2** |
| Ken is stressed out. He is fighting with Brad and there is a lot going on at mom's. Ken will see Ang this weekend. They will go to zoo with the grandkids on Sunday and see the show on Saturday. |
| **Summary - ChatGPT** |
| Ken discusses with Ang about feeling stressed due to work, family issues and too much going on. Ang offers support and asks if there's anything they can do to help. Ken expresses gratitude for listening to him vent. They discuss plans for the weekend, including a show on Saturday and seeing grandkids at the zoo on Sunday. They end the conversation by exchanging love and well wishes. |
| **Summary - InstructDS** |
| Ken is stressed because of work, fighting with Brad and too much going on at mom's. He will see a show on Saturday night and see the grandkids on Sunday at the zoo. |

Table 14: The second case study from SAMSum dataset and the generated summaries from different models.

| Case #1 from DREAM |
|---|

**Dialogue:**

W: What does the term "alternative energy source" mean?

M: When we think of energy or fuel for our homes and cars, we think of petroleum, a fossil fuel processed from oil removed from the ground, of which there is a limited supply. But alternative fuels can be many things. Wind, sun and water can all be used to create fuel.

W: Is the threat of running out of petroleum real?

M: It has taken thousands of years to create the natural stores of petroleum we have available at a much faster rate than it is being produced over time. The real controversy surrounding the amount of petroleum we have is how much we need to keep in reserve for future use. Most experts agree that by around 2025, the amount of petroleum we use will reach a peak. Then production and availability will begin to seriously decline. This is not to say there will be no petroleum at this point. But it'll become very difficult and therefore expensive to extract.

W: Is that the most important reason to develop alternative fuel and energy sources?

M: There are two very clear reasons to do so. One is that whether we have 60 or 600 years of fossil fuels left, we have to find other fuel sources eventually. So the sooner we start, the better off we will be. The other big argument is that when long time, which leads to some long-term negative effects, like global warming and the greenhouse effect.

**Queries:**

(1) What do we usually refer to when we talk about energy according to the man?

(2) What do most experts agree on according to the man?

(3) What does the man think we should do now?

**Summary - Human Reference**

(1) Fuel refined from oil extracted from underground.

(2) Oil production will begin to decline worldwide by 2025.

(3) Start developing alternative fuels.

**Summary - Alpaca**

(1) When we refer energy, according the Man, it usually refers to petrol, natural gas, and other fossils fuells. These fuell sources are limited and will eventually run out. Alternative fuelles are being developed to reduce the dependence on these limited fuellers.

(2) Most experTS agree THAT by AROUND

(3) The man thinks we must develop alternate energy and fuel source as soon as possible because it will help us reduce our dependence on fossils fuel and limit the negative environmental impacts caused by burning them.

**Summary - Flan-UL2**

(1) petroleum

(2) the amount of petroleum we use will reach a peak in 2025

(3) Find alternative energy sources

**Summary - ChatGPT**

(1) When we talk about energy according to the man, we usually refer to petroleum, a fossil fuel processed from oil removed from the ground, of which there is a limited supply.

(2) Most experts agree that by around 2025, the amount of petroleum we use will reach a peak and then production and availability will begin to seriously decline.

(3) There is no specific man mentioned in the given context who suggests something to do now. The conversation is about alternative energy sources and the need to develop them for a sustainable future.

**Summary - InstructDS**

(1) petroleum, a fossil fuel processed from oil removed from the ground, of which there is a limited supply

(2) that by around 2025, the amount of petroleum we use will reach a peak

(3) develop alternative fuel and energy sources

Table 15: One case study from DREAM dataset and the generated query-based summaries from different models.

| Case #1 from DialogSum |
|---|
| **Dialogue:** |
| Whitaker: Hello, so how are we feeling today?
Jill: Things are going well for me, doctor.
Whitaker: Am I correct in thinking that you are here for your annual physical?
Jill: Yes, I am applying for new health insurance, and I need a physical examination to qualify.
Whitaker: Your basic physical exam will include lungs, heart, blood levels, and eyes, ears, and nose.
Jill: I've been having a little trouble breathing. Would you look into that, please?
Whitaker: We can do an allergy test, and later I can send you for an asthma test.
Jill: I would appreciate it. When you give me a blood test, what are you looking for?
Whitaker: I am going to check your cholesterol, blood sugar, and white blood cell count.
Jill: I am expecting the tests to go well. I have been taking good care of myself. |
| **Summary - Human Reference** |
| Jill is going through an annual physical examination to apply for new health insurance, and Jill asks Whitaker to look into the breathing. |
| **Summary - BART** |
| #Person2# (Jill) comes to #Person1# (Whitaker) 's to have a physical examination for new health insurance and asks for an allergy test and an asthma test. |
| **Summary - Alpaca** |
| Doctor and patient discuss annual check-up, including labs, exam, insulin, asthmatic tests, chlolestrol, sugar and WBC count, allergies, healthy lifestyle, physical and mental health. |
| **Summary - Flan-UL2** |
| Jill is seeing Whitaker for her annual physical. She needs it to qualify for new health insurance. Jilll has been having trouble breathing. Whitakker will check her cholesterol, blood sugar, and white blood cell count. |
| **Summary - ChatGPT** |
| Jill visits Whitaker for her annual physical and requests for Asthma test due to breathing problems. Basic physical exam includes checking of lungs, heart, blood levels, eyes, ears, and nose. Blood test will include cholesterol, blood sugar, and white blood cell count. |
| **Summary - InstructDS** |
| Jill comes to Whitaker for Jill's annual physical because Jill needs a physical examination to qualify for new health insurance. Whitaker will do an allergy test and send Jill for an asthma test. |

Table 16: One case study from DialogSum dataset and the generated summaries from different models.

| Case #1 from TODSum |
|---|

**Dialogue:**

User: Hi, can you help me plan my Cambridge trip? I need a train from London Kings Cross to Cambridge.
Agent: I can help you with that. One leaves friday at 05:17, is that time okay for you?
User: Actually, I need to leave Tuesday after 10:30.
Agent: I have a train that leaves Tuesday at 11:17 that will arrive at 12:08. Would you like to make
    a reservation?
User: Yes, please make a reservation for 6 people. I'll need the reference number.
Agent: I was able to book your train. Your reference number is F13RS8CP. Is there anything else that
    you need?
User: I'm also looking for a place to eat. I would like an expensive south indian restaurant.
Agent: It looks like we don't have any of those. Would you like to try something else?
User: Hmm, are you sure? I want something south indian, expensive and in the centre of town.
Agent: I have curry garden for Indian in the centre of town, but no south indian.
User: What about one that serves mediterranean?
Agent: I have two options. I can recommend la mimosa. Would you like me to reserve a table?
User: You said you have two options. What is the other option available?
Agent: Other than the La Mimosa Restaurant, I also have the Shiraz Restaurant. Both are located in the
    center and are expensive. Which do you prefer?
User: Lets go with the Shiraz please
Agent: Ok perfect, would you like me to book a reservation for you?
User: No. I think I am ok although I would like the postcode.
Agent: The postcode is cb21dp. Is there anything else I can help you with today?
User: No, that is all. Thank you so much.
Agent: Let me know if you need anything else.
User: I don't need anything else. You were a great help. Goodbye!
Agent: Thank you for using our system!

**Summary - Human Reference**

the user wants the agent to help purchase 6 train tickets. this train leaves at 10:30 on tuesday, from london kings cross to cambridge. the user asks for the postcode of the restaurant. the restaurant is situated in the centre and has expensive mediterranean food.

**Summary - BART**

user wants the agent to help purchase 6 train tickets. this train leaves at 10:30 on friday, from london kings cross to cambridge. the user wants to book the restaurant seat for 6 people at 12:15 on tuesday. the restaurant is located in the centre and the food there is expensive.

**Summary - Alpaca**

The user needs to travel from Kings X to Cambrige and is looking to depart on a Friday at the latest. The agent is able offer a 5.15am train which is acceptable to the user. They also need accommodation for the night before and the agent suggests the Curry Garden for Indian food in Cambridge, which they find acceptable. Finally, the travel agent provides the necessary reference numbers for their reservations.

**Summary - Flan-UL2**

Agent helps the user book a train ticket from London Kings Cross to Cambridge. The user also wants to find an expensive restaurant in the centre of town.

**Summary - ChatGPT**

The user asks for assistance in planning a trip to Cambridge. The agent helps with booking a train and restaurant reservation with alternatives and provides reference numbers and postcodes. The user expresses gratitude and ends the conversation.

**Summary - InstructDS**

the user wants to book 6 train tickets. this train leaves at 10:30 on tuesday, from london kings cross to cambridge. the user asks for the postcode of the restaurant. the restaurant offers the expensive mediterranean dishes, which is located in the centre.

Table 17: One case study from TODSum dataset and the generated summaries from different models.