# OpenReview forum: "Instructive Dialogue Summarization with Query Aggregations"
_EMNLP/2023/Conference — EMNLP 2023 Main_

### Official Review · Reviewer_M5kz · 2023-08-04

**Soundness:** 4

**Excitement:**

4: Strong: This paper deepens the understanding of some phenomenon or lowers the barriers to an existing research direction.

**Paper Topic And Main Contributions:**

The paper aims to improve query-focused dialouge summarization models by leveraging synthetically generated query-dialouge-summary triples via Flan-T5-XL and use it to instruction-tune the same model (Flan-T5-XL). The authors propose a method to generate such synthetic data based on question-generation (query) and answer-generation (focused-summary) from reference summaries for a given dialouge. The authors curate synthetic data from three existing dialouge summarization datasets which help improve generalizability of final instruction-tuned model. Finally, authors also experiment with summary-length constraints in the instructions, and observe improvements over unconstrained version. With experiments, authors demonstrate that synthetic-data based instruction-tuning helps improve performance. The authors also support their analysis with human and ChatGPT-based evaluations along various subjective dimensions (e.g., fluency, faithfulness, informativeness, conciseness).

**Questions For The Authors:**

The following are suggestions which I think can help strengthen the paper further.

1. While I like the idea of conditioning on reference summaries, it could benefit the paper to formally investigate the quality of data generated when conditioning on the dialogues versus reference summaries. Currently, the paper makes the claim that it is challenging for models to generate data when conditioning on dialogues and empirical support would strengthen the paper.

2. QMSum is one of the primary work on query-focused summarization, which is of great importance given the nature of long meeting conversations and hence the value added by doing query-focused summarization over generic summarization. It would be interesting to see if the synthetic data generation approach could help improve performance on such a task of meeting summarization as well. The datasets currently investigated in the paper (SAMSum, TODSum, DialogSum) contain much shorter conversations (in terms of number of utterances per speaker and total number of utterances) compared to conversations in QMSum, where the benefits of query-focused summarization are much more pronounced. Such investigation can further help strengthen the claims of the paper and shed light on benefits of LLM-based synthetic data generation+instruction-tuning.

**Reasons To Accept:**

1. The study is thoroughly conducted with extensive number of summarization models, ablation studies and careful evaluations.
2. Query-focused summarization has many practical applications and the author's efforts in this direction are much desired.
3. I found it really interesting the way authors designed the synthetic data generation framework. For instance, conditioning on original reference summaries provided with existing datasets (instead of original dialouge) to curate the synthetic queries and focused-summaries is neat. As a result, the data generation process is slightly different from what the model is tasked to do at inference time, and hence the model is not generating data and making inferring for same task. As I understand, the model is generating questions and answers from a context (reference summary) which can then be recasted as a dataset for query-focused dialouge-summarization task.
4. The authors provide detailed human and ChatGPT-based evaluations to support their conclusions. Overall, the paper is also well written and easy to read.

**Reasons To Reject:**

I don't think I have any major concerns.

**Reproducibility:**

5: Could easily reproduce the results.

**Reviewer Confidence:**

5: Positive that my evaluation is correct. I read the paper very carefully and I am very familiar with related work.

**Typos Grammar Style And Presentation Improvements:**

Suggestions
1. Line 067. The introduction provides motivation for research on query-focused summarization in paragraphs 2-3. However, it suddenly switches to the challenge of instruction-based dialouge summarization in paragraph 4 (line 607). The flow could benefit by bridging this gap with one line motivating use of instruction-based training (e.g., Recent work has shown promising results with instruction-based training for many task [CITE], and has the potential to benefit query-focused summarization. However, the challenge [...] is scarcity of data.)

---

> ### Author Rebuttal · Authors · 2023-08-28
>
> We sincerely appreciate your thoughtful and thorough assessment of our paper on instructive dialogue summarization. Your positive feedback and recognition of the significance of our work are motivating. We are particularly gratified that you found our approach to synthetic data generation innovative and the design of our experiments valuable. We will definitely take your suggestions and make adjustments to further strengthen the paper:
>
> * **Comparison between Dialogue-Guided Query Generation (DQG) and Summary-Guided Query Generation (SQG)**: We agree with your recommendation to provide a more thorough comparison between DQG and SQG in our updated manuscript. Specifically, we have observed that DQG tends to generate superficial questions and focus on individual or limited utterances. In contrast, SQG leverages the concise information contained within human-written summaries, requiring a higher level of human reasoning and commonsense. Furthermore, DQG's performance is hindered by the inherent limitations of large language models in processing lengthy content, leading to suboptimal results. SQG queries, on the other hand, are formulated to be answerable from the summary itself, reducing uncertainty. We will expound upon the rationale behind our design choices and include a qualitative study to substantiate our claims.
>
> * **Extending from Daily Dialogue Summarization to Lengthy Meeting Summarization**: Your suggestion to explore the applicability of our approach to lengthy meeting summarization is well-received. We believe that our methodology can seamlessly extend to summarizing lengthy dialogues, thereby enhancing query-based summarization and dialogue comprehension. Our approach generates queries from reference summaries rather than relying on the entire lengthy dialogue. This approach avoids the challenges of using pre-trained language models for lengthy dialogue inputs. The major hurdle for meeting summarization lies in the limited availability of samples, which hinders its generalizability to different domains. Nevertheless, our method can contribute to mitigating data scarcity issues in the context of lengthy meeting summarization. We will introduce a dedicated discussion on this topic and considering the space limitation of the current manuscript, a thorough exploration might have to be left as future research. Meantime, more challenges will be encountered for meeting summarization and new solutions should be adopted.
>
>
> Thanks again for your thorough comments. Your insights will certainly guide us in further refining our work. We will improve the logic flow throughout the paper including the one regarding Line 067.

---

### Official Review · Reviewer_hwUr · 2023-08-04

**Soundness:** 3

**Excitement:**

3: Ambivalent: It has merits (e.g., it reports state-of-the-art results, the idea is nice), but there are key weaknesses (e.g., it describes incremental work), and it can significantly benefit from another round of revision. However, I won't object to accepting it if my co-reviewers champion it.

**Paper Topic And Main Contributions:**

The authors propose a comprehensive three-step approach for synthesizing high-quality query-based summarization triples. This process involves three key stages: summary-anchored query generation, query filtering, and query-based summary generation. To achieve this, they train a unified model named InstructDS (Instructive Dialogue Summarization) on three diverse summarization datasets containing multi-purpose instructive triples. To validate the effectiveness of the method, the authors conduct evaluations on four datasets, encompassing both dialogue summarization and dialogue reading comprehension tasks. Experimental results showcase the superiority of the approach over state-of-the-art models, including those with larger sizes. Moreover, the model exhibits higher generalizability and faithfulness, as corroborated by subjective evaluations conducted by human assessors.

**Reasons To Accept:**

The paper is well-written and motivated.
The paper introduces InstructDS, the dialogue summarization model that excels in following user instructions explicitly.
The method harnesses the question generation and answering capabilities of Large Language Models (LLMs).
Extensive evaluation of the proposed approach on three dialogue summarization datasets and one dialogue comprehension dataset shows that the results clearly illustrate a significant improvement compared to previous models.
Well-performed ablation study and case studies proves the efficacy of the approach.

**Reasons To Reject:**

The paper has limited technical contribution as the approach simply performs prompting LLMs.
Error analysis is missing.

**Reproducibility:**

4: Could mostly reproduce the results, but there may be some variation because of sample variance or minor variations in their interpretation of the protocol or method.

**Reviewer Confidence:**

4: Quite sure. I tried to check the important points carefully. It's unlikely, though conceivable, that I missed something that should affect my ratings.

---

> ### Author Rebuttal · Authors · 2023-08-28
>
> We sincerely appreciate your thoughtful assessment of our paper. Your recognition of the paper's clarity, motivation, design of InstructDS and experimental analysis is encouraging. We are committed to addressing your concerns regarding the technical contribution and error analysis of our proposed approach in a comprehensive manner:
>
> * **Problem significance**: It is important to emphasize the importance of instructive dialogue summarization. Unlike conventional dialogue summarization models, our approach can adapt to users' interests with flexibility. Previous attempts in news and meeting summarization have introduced new human-annotated datasets to tackle this problem, but they suffer from resource-intensive and scalability challenges. In our paper, we introduce a novel method for instructive dialogue summarization by synthesizing high-quality QDS triples and training a unified model. The InstructDS model demonstrates superior performance across four widely used datasets and contributes to enhanced dialogue understanding.
>
> * **Insights**: We acknowledge that simply prompting Large Language Models (LLMs) for data synthesis may not be as interesting or applicable to our scenario. In the context of InstructDS, the task demands a deep understanding of task properties and model capabilities. Prompting an LLM for instructive dialogue summarization is not straightforward. First, we generate queries by mining information directly from carefully crafted human summaries, which involve complex reasoning processes. These generated queries are less likely to be superficial or extractive from the dialogues, thus guiding the model towards better reasoning capabilities. Furthermore, we employ two innovative filtering approaches, text-based and semantic-based, to ensure quality and diversity in the collected QDS triples. The quality of these triples is verified by human experts. Finally, we train a unified model by aggregating all datasets and synthesized components.
>
> * **Strong evidence and extensive analysis**: We have conducted thorough experiments on four widely recognized datasets: SAMSum, DialogSum, TODSum, and DREAM. Our results outperform all existing models in these datasets. Additionally, we present an ablation study to demonstrate the effectiveness of each proposed component. To assess subjective quality, we utilize both human evaluation and ChatGPT evaluation. For error analysis, the ablation study provides strong evidence, and we include examples of synthesized QDS triples and model predictions in the appendix.
>
>
> Thanks again for your meticulous evaluation of our work. We would like to emphasize that instructive dialogue summarization is a non-trivial problem and our solution, which introduces synthesized triples based on a comprehensive understanding of various aspects, represents an effective approach to address this challenge.

---

### Official Review · Reviewer_YtPF · 2023-08-04

**Soundness:** 4

**Excitement:**

4: Strong: This paper deepens the understanding of some phenomenon or lowers the barriers to an existing research direction.

**Missing References:**

This paper should include the following citations to recent papers on query-focused summarization:

SumREN: Summarizing Reported Speech about Events in News. AAAI 2023

Exploring Neural Models for Query-Focused Summarization. NAACL 2022

Controllable Summarization with Constrained Markov Decision Process. TACL 2021

**Paper Topic And Main Contributions:**

This paper proposes a novel instruction-tuning method to expand the capability of dialogue summarization models. It devises a three-step pipeline approach to generate a large-scale dialogue summarization training dataset with the instruction demonstration style. Experiment results show that the proposed instruction tuning method helps PLMs outperform larger language models in various dialog understanding and summarization datasets.

**Questions For The Authors:**

Is there any difference between instructive dialogue summarization and query-focused summarization?

**Reasons To Accept:**

The proposed instruction tuning method helps PLMs outperform larger language models in various dialog understanding and summarization datasets.

The experiments in this paper are comprehensive. It manually verifies the quality of a subset of synthetically generated instruction data. It utilizes both n-gram-based metrics, LLM-based metrics, and human evaluation to evaluate the qualities of generated summaries. It also conducted a detailed ablation study to evaluate the contribution of synthetic QDF triples and length-aware augmentation.

The proposed method for generating instructional dialogue summarization data is novel and interesting. It helps alleviate the scarcity of query-based dialogue summarization.

**Reasons To Reject:**

It is not clear whether the proposed instruction tuning method can enhance the ability of PLMs to understand lengthy meeting dialogs. This paper should provide more discussion on this aspect.

The differences between instructive dialogue summarization and query-focused summarization are not clearly described. The authors should provide more discussions on the differences if they do not refer to the same concept.

**Reproducibility:**

4: Could mostly reproduce the results, but there may be some variation because of sample variance or minor variations in their interpretation of the protocol or method.

**Reviewer Confidence:**

4: Quite sure. I tried to check the important points carefully. It's unlikely, though conceivable, that I missed something that should affect my ratings.

---

> ### Author Rebuttal · Authors · 2023-08-28
>
> We want to express our appreciation for the thorough review of our paper and recognition of the satisfactory aspects of our work, such as its novelty, comprehensive analysis and performance improvements. We are also grateful for your insightful questions and we intend to address them point by point.
>
> * Can the proposed method enhance the ability of PLMs for understanding lengthy meeting dialogues?
>
>     * Certainly, our method can be directly applied to lengthy dialogue summarization datasets and improve the quality of query-based summarization and dialogue understanding. This capability arises from the fact that our proposed method is not constrained by dialogue length. To clarify, our approach involves the automatic generation of queries from reference summaries rather than replying on the lengthy dialogue itself. The filtering and query-based summary generation processes are all conditioned on the summary. This approach alleviates the challenges associated with using off-the-shelf PLMs to process lengthy dialogue inputs, which we have found to be suboptimal.
>
>     * It's worth noting that the focus of our paper is on daily dialogues, and we have conducted experiments using four popular datasets: SAMSum, DialogSum, TODSum, and DREAM. Regarding meeting summarization, as discussed in related work, we are aware of QMSum, which introduces a human-annotated query-based summarization dataset. However, meeting summarization datasets typically have limited samples due to the challenges of collection and transcription, which restricts their applicability to various meeting scenarios. Therefore, there is a pressing need for larger-scale meeting summarization datasets. Nevertheless, given the current circumstances, our proposed method can still alleviate the data scarcity issue.
>
> * Is there any distinction between instructive dialogue summarization and query-focused summarization?
>     * Yes and no. The term "instructive dialogue summarization" was introduced in our work. An ideal instructive dialogue summarization model should be capable of handling a wide range of instructions when generating summaries. As illustrated in Figure 2, our current model can accommodate general dialogue summarization, query-based dialogue summarization, and dialogue summarization with length control. We anticipate that the range of instructions will be expanded in future research, encompassing diverse sets of instructions and multi-round dialogue summarization scenarios. In the meantime, as a domain-specific model, we anticipate that instructive dialogue summarization could exhibit emergent capabilities as shown in general instruction-tuned LLMs.
>     * Concerning query-based summarization, most existing settings closely resemble dialogue question-answering. The summary is generated to address specific queries. However, this concept can also be enriched by considering various inquiries as queries. In this case, the concepts of instructive dialogue summarization and query-focused summarization become closely aligned. We will incorporate this discussion into the updated manuscript.
>
> * Missing references
>     * Thank you for pointing out the relevant papers. They will serve as supplementary references in addition to our current citations. We have already referred to some papers on controllable summaries and query-based summarization, and we will include the ones you've listed, along with necessary discussions.
>
> Once again, we extend our gratitude for your efforts and for recognizing the contributions of our work. Your comments are helpful in further improving the quality of our paper. By the way, please be aware of a caching issue in the 'Paper topic' section.

---

### Meta-Review · Area_Chair_1Snw · 2023-09-08

**Recommendation:** 4

**Metareview:**

The paper introduces the idea of instructive dialogue summarisation, which combines dialogue QA, general summarisation and query-focused summarisation into a single task. All reviewers thought: (1) this idea is novel and well-motivated; (2) the paper is well-written and easy to understand; (3) the experiments are extensive and thorough, including multiple datasets, summarisation models, automatic and human evaluation; and (4) the approach of training data generation via LLM is intuitive, creative, and impactful. There are no major concerns raised, although the paper could benefit from the following minor revisions: (1) clarifying the task (instructive dialogue summarisation) better, in particular how it is related/different to query-focused summarisation; (2) discuss why QMSum is not included as one of its datasets (seeing it's very relevant to the task); and (3) discuss or provide preliminary results on the impact of using dialogues vs. references for generating training data.

---

### Decision · Program_Chairs · 2023-10-07

**Decision:**

Accept-Main

**Comment:**

The paper introduces the idea of instructive dialogue summarisation, which combines dialogue QA, general summarisation and query-focused summarisation into a single task. All reviewers thought: (1) this idea is novel and well-motivated; (2) the paper is well-written and easy to understand; (3) the experiments are extensive and thorough, including multiple datasets, summarisation models, automatic and human evaluation; and (4) the approach of training data generation via LLM is intuitive, creative, and impactful. There are no major concerns raised, although the paper could benefit from the following minor revisions: (1) clarifying the task (instructive dialogue summarisation) better, in particular how it is related/different to query-focused summarisation; (2) discuss why QMSum is not included as one of its datasets (seeing it's very relevant to the task); and (3) discuss or provide preliminary results on the impact of using dialogues vs. references for generating training data.